# Fused Thermal and RGB Imagery for Robust Detection and Classification of Dynamic Objects in Mixed Datasets via Pre-Trained High-Level CNN

**Ravit Ben-Shoushan and Anna Brook***

Spectroscopy & Remote Sensing Laboratory, Spatial Analysis Research Center (UHCSISR),
Department of Geography and Environmental Studies, University of Haifa, Mount Carmel,
Haifa 3498838, Israel
* Correspondence: abrook@geo.haifa.ac.il

**Abstract:** Smart vehicles with embedded Autonomous Vehicle (AV) technologies are currently equipped with different types of mounted sensors, aiming to ensure safe movement for both passengers and other road users. The sensors' ability to capture and gather data to be synchronically interpreted by neural networks for a clear understanding of the surroundings is influenced by lighting conditions, such as natural lighting levels, artificial lighting effects, time of day, and various weather conditions, such as rain, fog, haze, and extreme temperatures. Such changing environmental conditions are also known as complex environments. In addition, the appearance of other road users is varied and relative to the vehicle's perspective; thus, the identification of features in a complex background is still a challenge. This paper presents a pre-processing method using multi-sensorial RGB and thermal camera data. The aim is to handle issues arising from the combined inputs of multiple sensors, such as data registration and value unification. Foreground refinement, followed by a novel statistical anomaly-based feature extraction prior to image fusion, is presented. The results met the AV challenges in CNN's classification. The reduction of the collected data and its variation level was achieved. The unified physical value contributed to the robustness of input data, providing a better perception of the surroundings under varied environmental conditions in mixed datasets for day and night images. The method presented uses fused images, robustly enriched with texture and feature depth and reduced dependency on lighting or environmental conditions, as an input for a CNN. The CNN was capable of extracting and classifying dynamic objects as vehicles and pedestrians from the complex background in both daylight and nightlight images.

**Keywords:** multi-sensors; anomaly fusion; pre-network fusion; physical value; complex environment; dynamic objects extraction

## 1. Introduction

Computer vision-based technologies are widely used in different applications, for example, in medical, agriculture, security, and conservation research. These achievements are also among the key pieces to many embedded Autonomous Vehicle (AV) applications [1,2] in the race toward developing a fully autonomous machine. However, it remains a challenging issue in the research community. Currently, smart vehicles with embedded AV technologies are equipped with multiple types of sensors aiming to gather data in different ranges that can be synchronically interpreted for a clear understanding of the surroundings (i.e., the locations of road users or other obstacles and the relative dynamic interactions between them). Recent machine learning developments have reached impressive achievements in computer vision tasks. A vast amount of research and methods have been developed for multi-sensor data fusion [3–6] to synergize the gathered information

while omitting redundant data. This allows for storing more knowledge and employing lower amounts of data.

Identifying features in a complex background often targets road users, i.e., people and vehicles, relative to the vehicle's perspective, whether static or dynamic. Scanning vehicles is dynamic in and of itself, whether they be static or moving, making the spatial representation of a complex environment time-dependent with no absolute static state. Each object's dynamic level is measured relative to the moving vehicle. Hence, challenges arise from several basic cumulative demands, including an ongoing and continuous sense of the vehicle's surroundings, dynamic driving abilities that fit different road types (highway, city roads, etc.), the ability to operate in different lighting and environmental conditions, and the real-time ability to detect static and dynamic targets (road users) in changing scenarios.

Lighting conditions affect the acquired data. Natural lighting level (daylight) is a dynamic factor, subject to the effects of the season and the time of day, and the sun's position and levels of radiation change throughout the day and seasons. The probability of blurring arises around dawn and sunset hours due to the sun's lower position. Less informative data samplings containing obscured features might occur due to the blurring effect resulting from the diversity of artificial night-time light sources, such as streetlights, vehicles' headlights, lighted signs, digital advertising screens, and building lights. Light frequencies generated by different types of light origins affect sensors differently. These days, LED lamps commonly used in the public domain in different lighting fixtures emit radiation in the range of visible light to near-infrared wavelengths (depending on the color of the installed LED lamp).

The sensors' ability to capture informative data is also influenced by weather conditions [7–10]. Rain, fog, and haze are masking effects caused by high amounts of water droplets, sand, and dust grains in the air. Cloudy or partially cloudy skies, extreme temperatures, and wet roads differently affect sensing capabilities along with the lighting conditions (natural or artificial lights). Shading, dazzling from objects with high reflectivity, or a shimmering effect might occur when puddles or hot roads are in the scanned scene. These many influences contribute to the high variability found in the acquired data. The additional influence could be exerted by the effect of trees and buildings' shadows, or for instance, the sudden darkness while driving through tunnels.

Evaluating the contribution of fused data to better perception by Convolutional Neural Networks (CNNs) is performed in relation to well-known CNN object detectors trained with RGB images and considering the suitability of the model's architecture to operate in real-time with low computational resources. Among the CNN algorithms for image recognition tasks are two-stage-based architectures, e.g., AlexNet, VGG16, GoogLeNet, and ResNet. Region-based CNN (R-CNN) [11] proposed a bounding-box regression-based approach, later developed into Fast R-CNN [12], Faster R-CNN [13], and R-FCN (Region-based Fully Convolutional Network) [14]. Faster R-CNN uses the region proposal network (RPN) method to classify bounding boxes, followed by finetuning to process the bounding boxes [15,16]. One significant drawback of the two-stage architecture is its slow speed detection, resulting in the inability to produce real-time results as required for AV applications. An alternative approach is a one-pass regression of class probabilities and bounding box locations, e.g., Single Shot Multibox Detector (SSD) [17], Deeply Supervised Object Detector (DSOD) [18], RetinaNet [19], EfficientNet [20], You Only Look Once (YOLO) architecture developed by Redmon et al. [21], etc. These methods unite target classification and localization into a regression problem, do not require RPN, and directly perform regression to detect targets in the image.

YOLO became a widely used algorithm due to the model's small size and fast calculation speed. It constructs a backbone for pre-training and a one-stage head to predict classes and bounding box (dense prediction) layers. Subsequent versions described as YOLO V2 [22], YOLO V3 [23], YOLO V4 [24], and YOLO V5 [25], published in the following years, attempted to improve the low-detection accuracy of the original model and its

inefficiency in small target detection. The main developments in YOLO versions were reviewed by Jiang et al. [26]. YOLO V2 offered better and faster results by improving the inaccuracy positioning, lowering the recall rate, and switching the primary network used for training from GoogLeNet to Darknet-19, simplifying the network's architecture. In YOLO V3, feature graphs of three scales were adopted using three prior boxes for each position, later divided into three scale feature maps added to a multi-scale detection. However, the feature extraction network used the residual model, which contained 53 convolution layers (Darknet-53) instead of the Darknet-19 used in YOLO V2, enabling it to focus on comparing data. YOLO V4 optimized the speed and accuracy of object detection. Some of its substantial improvements include adding spatial pyramid pooling (SPP) block with an increased receptive field, which separates significant features, MISH activation function, Cross-Stage-Partial-connections (CSP), enhancement by mosaic data augmentation, and Generalized Intersection over Union (GIOU) loss function. YOLO V5 is similar to YOLO V4, but it is based on the PyTorch platform, different from Darknet, which is mainly written in C.

The neural networks fed with RGB and thermal data gained tremendous progress in the last decade with dozens of algorithms, offering a variety of methods for the fusion of image sources in different phases of the learning process [27–31]. The fused data might overcome the challenges of accurate detection of dynamic objects in a complex scene captured by dual sensors on moving vehicles. Complex scenes relate to changing environmental conditions in daylight or nightlight. Changing lighting effects causes different reactions to dynamic scenes.

In a review of real-time detection and localization algorithms for AV conducted by Lu et al. [32], the authors concluded that since no single sensor can meet all localization requirements for autonomous driving, fusion-based techniques would be the research focus for achieving a cost-efficient self-localization for AV. In addition, they pointed out that future research is required to focus on sensors' faulty detection and identification techniques and imperfect data modeling approaches to ensure robust and consistent AV localization. Chen et al. [33] summarized the importance and advantages of visual multi-sensor fusion. In a review of sensing systems for AV environmental perception technologies, a set of open challenges were listed: the lack of a theoretical framework for targeting generic fusion rather than specific fields; ambiguity in associating different sensors' data; poor robustness; insufficient integration of fusion methods; and the lack of a unified standard specification and evaluation criteria.

Many reviews have been published in recent years on infrared and visible image fusion methods in the context of AV [6,28,30,34]. The fusion process in the context of neural network architectures may occur in different stages of the learning process: pre-network fusion generates a new single input for the network using by fusing the row data; network-based fusion (or fusion as part of multimodal architecture) is categorized by the phase in which the data is fused [35]. The early fusion approach uses multiple origins of raw data as input. Data from each origin is separately processed to unite and refine data from the different sensors, followed by a pixel-level fusion layer. The middle (halfway) fusion processes each input layer using parallel convolution-based encoder blocks to extract the valuable data from each source. The extracted features are then fused and forwarded as a single input or additive data (e.g., feature maps, optical flow, density maps, etc.) for the next network block to be interpreted. Li and Wu [36] proposed fusing feature maps of VIS and IR images that were decomposed using an encoder consisting of a convolutional layer and a dense block prior to reconstructing the fused data with a decoder block. The late fusion (model level) selectively takes place after the network has separately segmented and classified features from each data source based on pre-defined thresholds and the situation in a test.

The main aim of this paper is to evaluate the best method to pre-process the fusion of multi-sensorial data (RGB and thermal cameras) captured using sensors in motion

(mounted on AV) that enables CNNs to robustly detect and classify vehicles and pedestrians in complex backgrounds and mixed datasets (daylight and nightlight images).

The developed method includes a pre-processing stage of data fusion combining anomaly detection, enabling the classification of dynamic objects from the complex background. A novel anomaly-based feature extraction process is proposed to overcome the above-mentioned AV challenges in CNN classification tasks. The RGB images are transformed to the intensity, hue, and saturation coordinates, and the improved contrast image is registered to the thermal (IR) image via the affine module to enhance and generalize the RGB images. A global Reed–Xiaoli anomaly detector map (GRXD) from the enhanced RGB data is calculated and normalized to an anomaly image representation. Then both enhanced and anomaly images are integrated with the IR image into a new physical value image representation. The fused images are robustly enriched with texture and feature depth, reducing dependency on lighting or environmental conditions. Such images are used as input for a CNN to extract and classify vehicles and pedestrians in daylight and nightlight images.

The rest of this paper is organized as follows. Section 2 describes levels of fusion, different methods to decompose the raw images prior to the fusion process, and fusion rules to combine the decomposed images. In addition, the challenges of fusing images from different sensors are described. Section 3 describes the FLIR dataset and the tested scenarios, as well as the proposed pre-processing of the dual data followed by feature extraction methods (range filter and RXD anomaly detection) and different data fusion techniques. Thereafter, the setting and training of CNN YOLO V5 and the final dataset versions used to train the networks are presented. In Section 4, the classification results of the trained networks are detailed. Further, in Section 5, the results and the contribution of the proposed method to reduce network failure in detection and classification tasks are discussed. Finally, the conclusions are presented in Section 6.

## 2. Related Works

In image processing, it is common to relate to three levels in which the data can be fused: Pixel-level fusion, also known as low level, where the raw pixel data of both images are fused [37]; Feature-level (region-based) fusion, which implies that the source images are first separately processed to extract the features of interest based on mutual and distinct characteristics with the extracted features then used in the fusion process; Decision-level (high-level fusion), where each source is initially processed and understood. At this level, the fusion only takes place if the extracted and labeled data meets the predetermined criteria. Most pixel-level fusion methods are based on the multi-scale transform in which original images can be decomposed into components of different scales by means of low-pass and high-pass filters. Multi-scale-transform fusion schemes consist of two key steps: the multi-scale decomposition method and fusion rules. Then, a corresponding inverse multi-scale transform is applied to reconstruct images using coefficients. Pyramid transform [34] decomposes sub-images via a pyramid structure generated from different scales of spatial frequency (e.g., Laplacian pyramid transform and Steerable decomposition technique). The Wavelet Transform proposed by Mallat [38] is a fast and efficient method for representing multi-scale uncorrelated coefficients and is widely used in fusing visual and thermal images. Discrete Wavelet Transform (DWT) decomposes the source image signals into a series of sub-images of high and low frequencies at zero scale space representing the detailed coefficients and approximation coefficients. The approximation coefficients can be further decomposed to the next level (to detail and approximation) repeatedly until the desired scale is reached. Regardless of its robustness, the DWT is known to suffer oscillation problems, shift variance, aliasing, and lack of directionality. Stationary Wavelet Transform (SWT) solves the problem of shift-invariance, thus contributing to preserving more detailed information in the decomposition coefficients [39]. A dual-tree complex Wavelet Transform shows improved performance in computational efficiency, near shift-invariance, and directional selectivity due to a separable filter bank [40]. Lifting Wavelet

Transform has the advantages of adaptive design, irregular sampling, and integral transform over DWT [41]. Additional techniques include lifting Stationary Wavelet Transform [42], redundant-lifting non-separable Wavelet multi-directional analysis [43], spectral graph Wavelet Transforms [44], quaternion Wavelet Transform, motion-compensated Wavelet Transform, multi-Wavelet, and other fusion methods being applied at the feature level due to their spatial characteristics. Gao et al. [45] used the non-subsampled contourlet transform (NSCT) for its flexibility and for being fully shift-invariant. The edge-preserving filter technique was combined with the fusion method [46]. This technique aims to decompose the source image into a smooth-base layer and one or more detail layers. As a result, the spatial consistency of structures is preserved while reducing halo artifacts around the edges.

The fusion rules set the method to combine the decomposed coefficients, such as coefficient combination (max and weighted averages) in pixel-level fusion. When fusion takes place at the feature level, fusion rules are according to the region level. The most representative method for feature level is based on the salient region, which aims to identify regions more salient than their neighbors. Other fusion rules are sparse representation-based methods that aim to learn an over-complete dictionary from a large amount of high-quality natural images. Each source image is decomposed into overlapping patches using a sliding window strategy. Furthermore, an over-complete dictionary is learned from many high-quality natural images, and sparse coding is performed on each patch to obtain the sparse representation coefficient using the learned over-complete dictionary. The fusion is applied according to the given fusion rule, reconstructing the image according to the fusion coefficients and the learned over-complete dictionary. This method can enhance the fused images to a meaningful and stable representation, reduce visual artifacts, and improve robustness.

The feature-level fusion process aims to identify objects by their regional characteristics. Accurately segmenting the target object's foreground from its background is a key phase for better object detection. Many algorithms were proposed to segment the information from the visualized data for AV purposes, as well as for medical procedures and early disease detection, smart agriculture, defense and security purposes, and many other fields of research. The color transformation is used in various disciplines in the pre-processing phase of data fusion, before segmentation and classification tasks, to mainly enhance the feature's border without blurring the featured foreground. Saba et al. [47] used Laplacian filtering followed by HSV color transformation to enhance the border contrast of images of skin lesions as part of pre-processing before color CNN-based segmentation and detection of melanoma. Afza et al. [48] used HSI transformation to enrich the contrast of video frames before fusion-based feature selection to target human action recognition. Adeel et al. [49] applied lab color transformation before multiple feature fusion tasks guided by the canonical correlation analysis (CCA) approach to recognize grape leaf diseases.

In the AV context, the data can sometimes be treated as features, a saliency map, or optical flow extraction. Images captured by sensors in motion contain measured characteristics by time. Therefore, the segmentation of the complex foreground (i.e., object movement in space) from the complex background (also due to unexpected environmental effects) is still considered a challenging task.

Morphological-based approaches, such as texture, color, intensity level, or shape-based methods, for example, can be separately and selectively used on each of the source images to extract the object of interest. An additional approach is to enhance the object's border and, thus, segment it from its background. Researchers offer various techniques, morphological and statistical, to straighten the object's boundaries and suppress its background, or a combination of both, for better feature extraction before the fusion process. Following the segmentation task, extracted layers (of pre-processed foreground and background) can be fused using a pixel fusion method based on their regional properties.

The fusion of contradictory signals might cause destructive interference. Therefore, finding the ultimate color coordinate representation yielding the most informative fused data is of great importance, more so when targeting a robust pre-processing for images from different scenarios, lighting, and weather conditions. Mustafa et al. [31] designed a self-attention mechanism combining multi-contextual and complementary features of IR and RGB images into a compact fused image representation.

However, a prerequisite for successful image fusion arises when using images from different sources: images should be strictly aligned in advance. Data acquisition using multiple sensors for AV is considered a common and acceptable method. Each sensor lens' parameters and relative position result in different information being captured. Image registration is the process of adjustment between two images captured by a single sensor at different times or by two (or more) sensors from different angles. The registered image is aligned with the same coordinate system as the original image through a transformation of the registered image matrix. Precise and accurate image registration is necessary for accurate object detection [50–54].

## 3. Methodology

### 3.1. Datasets

The existence of large and varied datasets is a cornerstone for the generic learning process. In many previous works on different aspects of AV, the availability of suitable datasets for training and testing was discussed [50,55,56]. Recently, Ellmauthaler et al. [53] presented an RGB and IR video database (VLIRVDIF), encouraged by the shortage of publicly available RGB-IR-synchronized dual databases. The authors also proposed a registration method to align the dual sensor data taken in distinct recording locations with varying scene content and lighting conditions. However, the offered dataset was captured by fixed sensors that were pre-calibrated at each location. The targeted dataset of visual and thermal multi-sensors synchronized and annotated real-world video captured from moving vehicles was found to be almost unavailable. In addition, the various scenarios and the changing environmental conditions' representation makes the datasets even harder to obtain.

In July 2018, FLIR Systems, Inc. released an IR dataset for Advanced Driver Assist Systems (ADAS) [57]. The dataset contains over 14K images of daylight and nightlight scenarios, acquired via synced RGB and IR cameras mounted on a vehicle while driving in Santa Barbara, California. The captured scenes correspond to urban streets and highways between November and May with clear-to-overcast weather. The IR images were recorded using FLIR Tau2 640 × 512, 13 mm f/1.0 (HFOV 45°, VFOV 37°) and FLIR Black-Fly (BFS-U3-51S5C-C) 1280 × 1024, Computer 4–8 mm f/1.4–16-megapixel lens for RGB images. The centerline of the images was approximately located 2 inches apart and collimated to minimize parallax. The dataset was recorded at 30 Hz. Dataset sequences were sampled at 2 frames/sec or 1 frame/sec. Video annotations were performed at 30 frames/sec recording (on IR images). Cars, as well as other vehicles, people, bicycles, and dog classes, were annotated. Since its publication, the FLIR dataset has been used in many research works to detect objects in adverse weather conditions using thermal images either as the main goal or as complementary data for other sensors' data extraction [58–60].

The FLIR-ADAS dual dataset was chosen to train, test, and validate the proposed model. About 2500 diverse images from the FLIR-ADAS dataset were used to train, validate, and test networks for data pre-processing phase evaluation. The RGB image dimensions are 1600 × 1800 × 3 (Figure 1a) and 1536 × 2048 × 3 (Figure 1c). All dual IR image sizes are 512 × 640 (Figure 1b, Figure 1d).

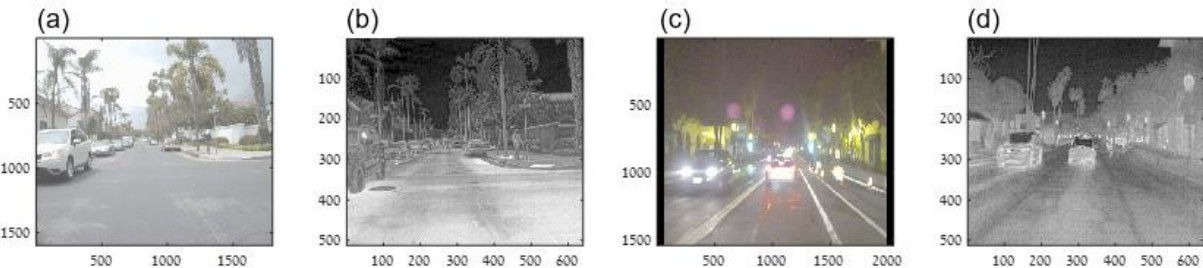

**Figure 1.** Example of FLIR dataset dual image: (**a**) FLIR_02767 RGB image size 1600 × 1800 × 3; (**b**) FLIR_02767 IR image size 512 × 640; (**c**) FLIR_05563 RGB image size 1536 × 2048 × 3; (**d**) FLIR_05563 IR image size 512 × 640.

The dataset included: near and far objects; diverse scene representation, such as main and side urban roads, city junctions, and intercity highways; different conditions under clear daylight, such as sunny sky, cloudy sky, dazzling low sun of twilight hours in front and from behind the camera, object under shadowed area, etc.; and nightlight with low and strong street lighting, and dazzling objects. Examples of diverse scenes are shown in Figure 2.

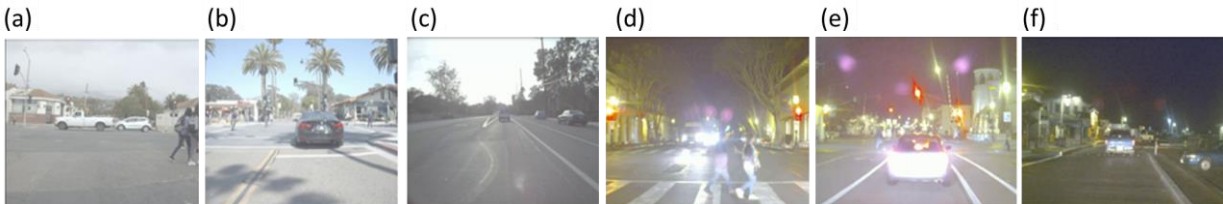

**Figure 2.** Diverse scenes, lighting conditions, and different road users. (**a–c**): Day images are (**a**) a cloudy sky, a city junction, a and mixed scene (pedestrians, cars); (**b**) a clear sky, a city junction, and a mixed scene (pedestrians, cars); and (**c**) a dazzling low-sun, intercity highway. d–f: Night images are (**d**) dazzling lights, a city junction, and a mixed scene (partially lighted pedestrians, cars); (**e**) backscatter flashing; and (**f**) an urban street with low luminance.

The final dataset contains 1275 unique images of natural luminance conditions (daylight images) and 1170 unique images of artificial lighting conditions (nightlight images) of diverse scenes, comprising various sources of lights and objects of different scales and appearances, as described above. The annotation ratio is 19% pedestrians and 75% cars. The remaining 6%, consisting of bicycles and pets, were ignored due to their low representation. Daylight and nightlight images were grouped into a dataset named Mixed Dataset, containing a total of 2445 images.

### 3.2. *Analysis*

Analysis workflow consisted of three steps: pre-processing, processing, and post-processing.

#### 3.2.1. Pre-Processing

The pre-processing workflow is presented in Figure 3 and contains the following steps: color transformation for the RGB image data; image registration (applied on RGB data according to IR); anomaly-based feature extraction; new RGB representation/reconstruction; and pixel-level and feature-level based fusion.

The proposed pre-processing workflow is as follows.

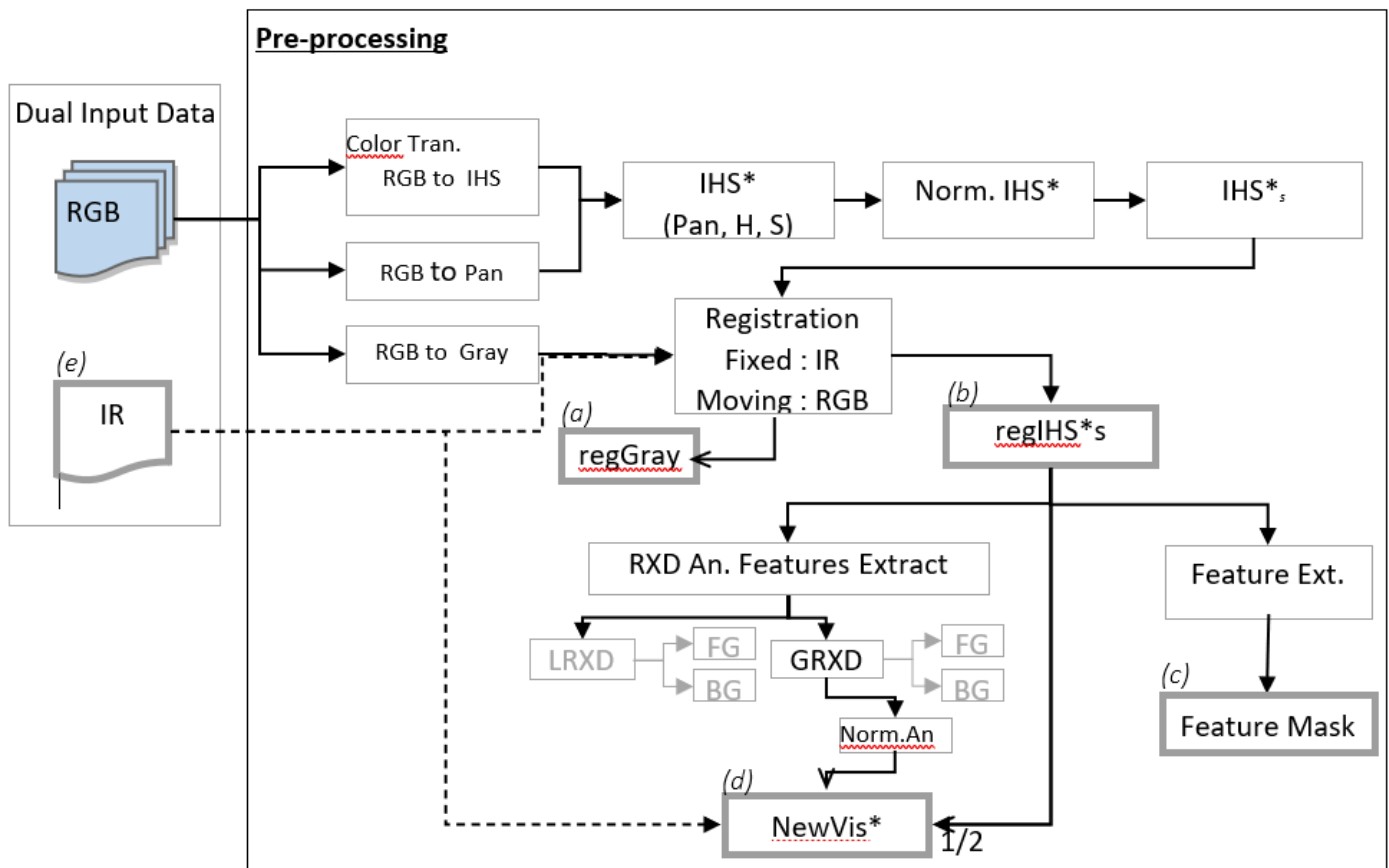

**Figure 3.** Schema of the proposed pre-processing workflow. Final products of the pre-processing phase are a registered grey representation of an RGB image (**a**), a registered saturation layer of an IHS image (**b**), a feature mask based on the (**b**) layer (**c**), and a new visual image representation (**d**) constructed from the integration of (**b**), RXD-based global anomaly of (**b**) and IR image (**e**).

Color Transformation

The preferred color transformation was tested according to linear methods, e.g., grey images, HSV, LIN, YUV, YIQ, YCbCr, and non-linear methods, such as various IHS color spaces. In linear color transformations, the color is created by splitting the reflected radiation from each object into three parameters: intensity (the amount of reflected light), usually scaled (0–1), and the range from absolute white to absolute black. The other two parameters represent chromatic content. In non-linear color transformations, three different components (intensity, hue, and saturation) are used for the color spatial representation. Intensity is also known as value or luminance. Al-Wassai et al. [61] compared different models of intensity, hue, and saturation (IHS) color spaces.

In this study, each transformed band was split into its component layers to measure the most informative data of the RGB image in a single-layer representation. The color transform algorithm was defined as follows: RGB image was reshaped from a 3D to a 2D matrix. The dot product was calculated from the reshaped RGB image with the IHS transform of the cylinder structure, which was described by Carper et al. [62] and has been widely implemented since then [63]. The axes were rotated to $L\alpha^*\beta^*$, a non-linear color space. The transformed IHS layers were rearranged into a matrix according to the original input RGB dimensions. Next, the three transformed layers were concatenated at the third dimension to reconstruct the new image. Following Al-Wassai et al.'s [61] method, the intensity (L) band is replaced with the panchromatic image. A panchromatic image representation of an RGB image is created by converting it to grayscale with the portions given in Equation (1).

$$I_{Gray} = 0.299 \times I_R + 0.587 \times I_G + 0.114 \times I_B \tag{1}$$

where IGrey is the grey representation of RGB image, IR, IG, and IB corresponding to the red, green, and blue channels, respectively. Final layers used to concatenate the new image correspond to IGray, H, S. where H and S are the hue and saturation layers of the IHS transformed bands. The new image values were then normalized by rescaling to the interval [0–1], multiplying by 255, and setting to UINT8 data type.

Registration

The different methods relate to the type of geometric adjustment to apply to the matrix's values include: non-reflective similarity, similarity, affine, and projective. In their work, Jana et al. [64] used affine transformations to learn the distortions caused by camera angle variations. Li et al. [65] applied an affine transformation to feature-wise edge incorporation as an initial process to EC-CNN for thermal image semantic segmentation. In this work, the transformation matrix was calculated according to fixed points (IR image) and moving points (RGB image), that were manually registered. The dual captured data in the FLIR dataset offers images that were extracted from several recording sessions. Nevertheless, being mounted on a vehicle, the captured images reflect minor shifts in the sensor's position, resulting in a non registrated images. Since the misregistration is minor (shift of 2-3 pixels) a manual registration approach was proposed. For each recording session, a single t-form for the entire session's corresponding dual images, was calculated. The residual misregistration (subpixel level) was was further included in that general inaccuracy caused by the vehicles' inherent shift and treated via data fusion. The coefficients matrix was based on affine translation. The saturation layer of the IHS transformed image was registered to the size and coordinates of IR image. This was performed using a predefined t-concord matrix suited to each pair of dual images,. The output layer was named regIHS*s. In addition, the grayscale image of RGB was registered to the IR image. The output was named regGray. These layers will be used in the next steps.

Features Extraction

Pedestrians and cars are objects with great diversity in texture, shape, and color. Therefore, threshold-based feature extraction may enhance object detection by robustly synergizing RGB and IR images. Naturally, setting a single value as an intensity-level threshold on a mixed dataset will serve for extracting different patches in daylight and nightlight images with respect to the following aspects: the surrounding temperature, the scene's brightness, the object's relative heat emission, its surface's texture, and its relative dynamics. In addition, the fusion of salient features (extracted from the IR image) with areas of high-level intensity (as in dazzled areas in nightlight RGB images) may result in low contrast between the fused object and its background. Thus, masking the RGB blurring patches or reducing their intensity might enhance contrast. Challenges in robustly extracting features from daylight and nightlight images using data from RGB and IR are shown in (Figure 4). Four captured scenes are presented: dual-RGB and -IR images (Figure 4a,g,m,s) and (Figure 4b,h,n,t) corresponding to daylight (Figure 4a,g) and (Figure 4m,s) to nightlight. Blue bounding boxes were added to mark car objects as a region of interest, and magenta bounding boxes were used to mark pedestrians. A general threshold of 0.7 was set to create a foreground binary mask of the IR image (Figure 4c,i,o,u). The negative mask exhibits the segmented background (Figure 4d,j,p,v). A binary mask of IR daylight images segments the two walking figures and road surface as the foreground (Figure 4c) and the marked cars as the background. In Figure 4i, the same threshold level of the IR image was used to extract the two figures on the sidewalk (vegetation background, shaded sidewalk), which were difficult to distinguish on the RGB image. However, it failed to extract the figure on the road (Figure 4i, left box), segmenting it as background. Figure 4e,k,q,w shows a segmented background using the IR mask applied to the RGB

image. Fusion of the IR salience features with masked RGB as background (Figure 4f,l,r,x) resulted in enhancing car contrast while reducing pedestrian contrast (Figure 4f). These challenges occurred for nightlight images as well as for fused features of the low contrast to the background (crossing pedestrians in Figure 4r). The mixed effect of fused features due to the complex background is shown in (Figure 4x).

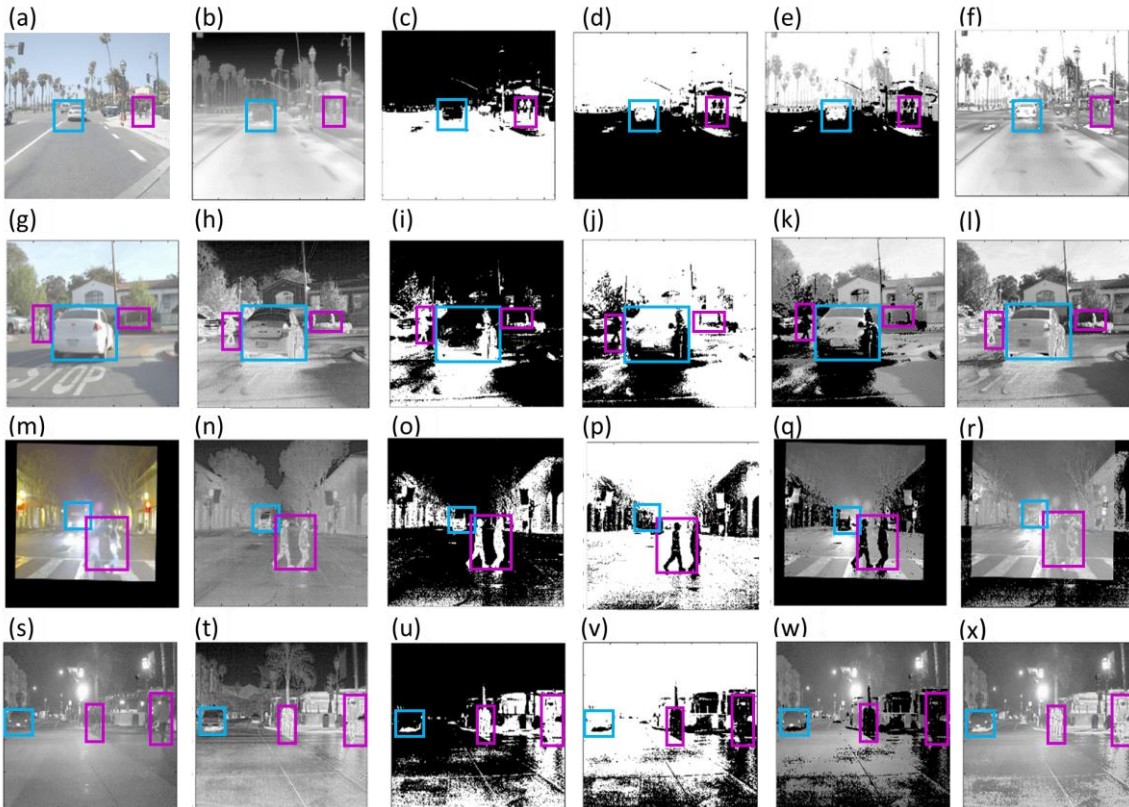

**Figure 4.** Rows: image samples from the test dataset. Daylight images: FLIR _02163 (**a**–**f**) and FLIR _05011 (**g**–**l**); Nightlight images: FLIR_05697 (**m**–**r**) and FLIR_08926 (**s**–**x**). Columns: registered RGB images (**a**,**g**,**m**,**s**); original IR image (**b**,**h**,**n**,**t**); a 0.7 threshold binary mask of the IR image (**c**,**i**,**o**,**u**); negative of the IR binary mask as background binary mask (**d**,**j**,**p**,**v**); masked registered RGB image as scene's background (**e**,**k**,**q**,**w**); the fusion of masked RGB background and masked IR foreground (**f**,**l**,**r**,**x**). The blue bounding boxes were added to point out car objects, and magenta BBs were added to mark pedestrians.

Aiming to straighten the object's boundaries, a 3 by 3 Range Filter (RF) was applied to the *regGray* image: The RF output matrix represents the intensity range (min–max) of each pixel's neighbor. Then, morphological operations (dilation and erosion of 3 × 3 structure element) were applied to only capture the feature's boundaries and suppress the background's texture. A binary mask was calculated, and a small-feature elimination was applied to reduce noise. The created feature mask (Figure 5a–d) was then used in the fusion process.

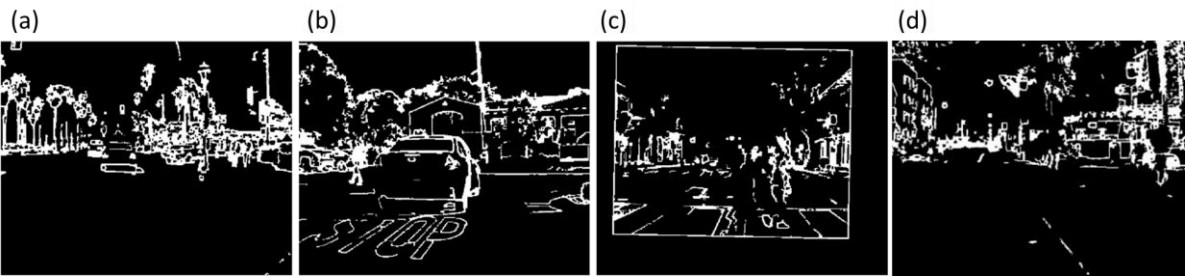

**Figure 5.** RF_VisMask. daylight image FLIR_02163 (**a**); daylight image FLIR _05011 (**b**); nightlight image FLIR _05697 (**c**); and nightlight image FLIR_08926 (**d**).

Proposed Anomaly-Based Pre-Process

Anomaly Detection (AD) methods, e.g., BACON [66] and RXD [67], are statistical approaches to measuring each pixel's probability of belonging to the background, assuming a multivariate normal distribution of the background. Guo, Pu, and Cheng [68] examined several methods to detect anomalies. In the pre-processing phase, using AD for feature extraction can contribute to the generalization of the data. While analyzing RGB images, a global anomaly of daylight images contributes to better foreground segmentation, but anomaly-based foreground extraction of nightlight images tends to extract blurred patches.

The RXD is a commonly used method for anomaly detection. In this method, no specific data is marked as an anomaly. It relies on the assumption that the image background is multidimensionally distributed. Therefore, the background pixels' sampling will have a lower probability value, and the anomalies are expected to have a higher value of probability. Based on this assumption, the local anomaly of an image is calculated by Equation (2),

$$RXD_{(x)} = (x - \mu)^T \textstyle\sum^{-1}(x - \mu) \tag{2}$$

where $RXD_{(x)}$ stands for the image's local anomaly, $(x - \mu)^T$ is the transposed vector of values calculated by subtracting the $(x)$ pixel in the test from $\mu$ (the mean of x's 8-pixel neighborhood) and $\sum$ is the covariance matrix usually deployed as the image covariance instead of the neighborhood covariance. A global anomaly will be calculated according to Equation (3),

$$GRXD = (X - \mu_G)^T \textstyle\sum_G^{-1}(X - \mu_G) \tag{3}$$

where $X$ is the pixel under testing (or a vector of the sliding window values) and $\mu_{(G)}$ and $\sum_G$ are the mean and covariance of all pixels in the image, respectively. The expression $(X - \mu_{(G)})^T$ represents the transposed substruction vector. This expression is later multiplied, firstly by the image covariance powered by −1 and secondly by $X - \mu_{(G)}$. An example of the local and global anomaly of an image is shown in Figure 6.

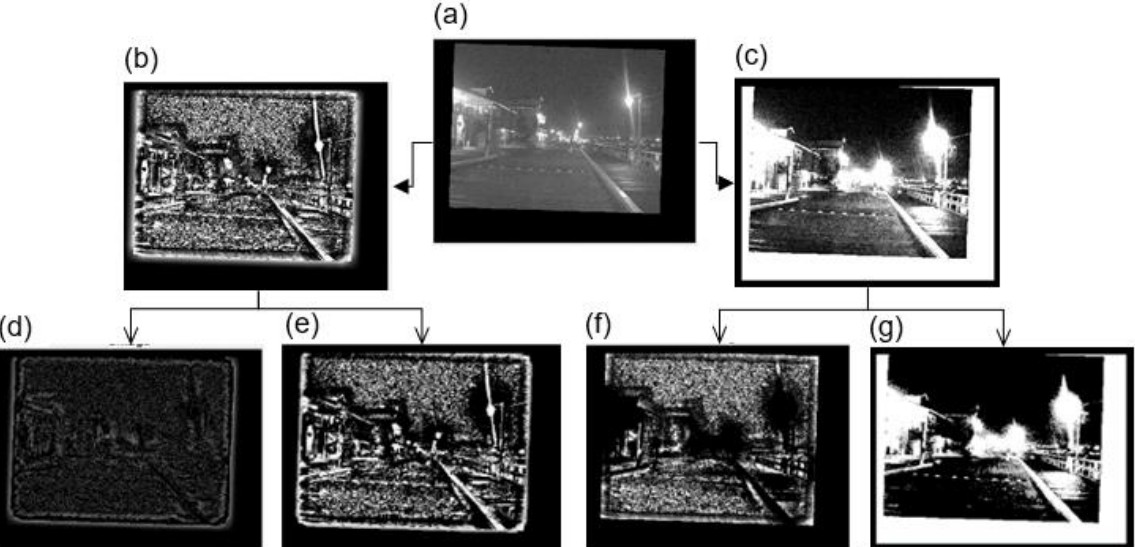

**Figure 6.** Extracting night image FLIR_06001 (**a**) to the local anomaly (**b**) and global anomaly (**c**). Both local and global anomaly images can be segmented into the background, (**d**) and (**f**), and foreground, (**e**) and (**g**).

Following the pre-process, as detailed earlier (color transform and image registration), the saturation layer of the *regIHS\*s* is used to calculate the image's global anomaly according to Equation (3). Next, *GRXD* is normalized between 0 and 1 and further multiplied by 255.

A pseudo-RGB (*NewVis\**) image is reconstructed according to Equation (4). Given that *regIHS\*s* is a registered, color-converted 2D layer of the RGB image, *normAN* is the normalized detected global anomaly of *regIHS\*s*, and *IR* is the original IR image.

$$NewVis^* = \frac{regIHS^*s}{2} - normAN + IR \tag{4}$$

The *NewVis\** is calculated by subtracting the normalized anomaly values from half of the transformed RGB layer and adding IR values. This image reconstruction aims to narrow the diversification of digital representation caused by natural differences between daylight and nightlight images, hence creating a generalized RGB representation.

### 3.2.2. Processing

### Image Fusion Methods

The Stationary Wavelet Transform (SWT) algorithm is used for the fusion process as demonstrated in [69,70]. SWT decomposes the input signals into scaling and Wavelet coefficients, enabling the preservation of the image texture and edge information while reconstructing the fused signals from the sub-bands back to the image. By being shift-invariant, SWT can effectively reduce distortion caused by the heterogenous data representation of RGB and IR images.

In the SIDWT decomposition phase, each row in the image is separately filtered using high-pass (HP) and low-pass (LP) filters. Next, this image is filtered again along the columns. The output is four sub-bands in the first decomposition level. Three sub-bands (LH, HL, and HH), also known as filter coefficients, contain the horizontal, vertical, and diagonal frequencies' details along with sub-band LL. The approximation data are transferred onto the next decomposition level. The decompose frequency is increased by a factor of 2(i-1) on the $i_{th}$ level of the algorithm, so each n decomposition level will have 3n+1 sub-bands. In this paper, a SymLet Wavelet (sym2) is applied. Following the pre-processing steps in Section 3.2, the fusion process is presented in Figure 7.

The fusion process is shown below.

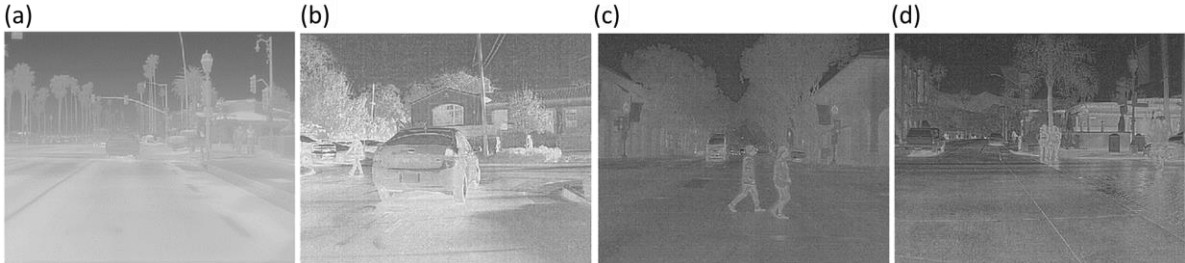

**Figure 7.** Fusion process schema and the resulting datasets for training, validating, and testing the networks.

Each unique image from the final dataset (2445 images) was processed according to the proposed algorithms described in Section 3.2, resulting in three types of processes for comparison: dual-RGB (*regIHS*s*) and -IR images fused at the pixel level, range-filter-based feature extraction of the pre-processed RGB image (Feature Mask) fused with IR images (as RF Feature Fusion), RXD-based anomaly feature extraction of pseudo-RGB (*NewVis**), and IR image fusion (as RXD Anomaly Feature Fusion). The original IR images dataset (as IR) was also tested for comparison vs. dual data. Examples of feature fusion images are shown in Figure 8. The created *NewVis** fused with the corresponding IR image.

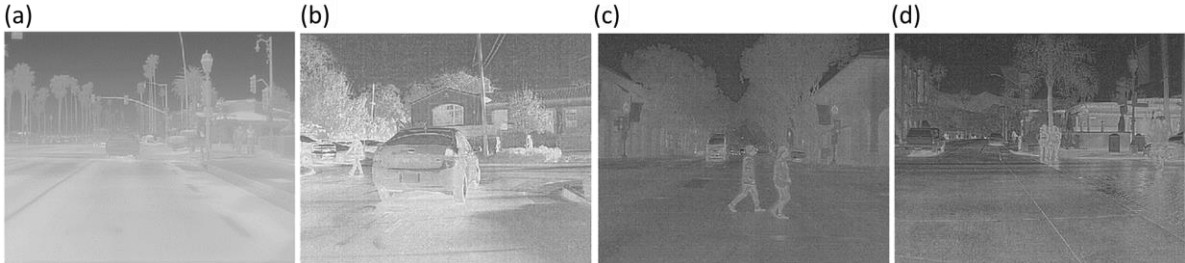

**Figure 8.** Fused images from feature fusion DS: daylight image FLIR_02163 (**a**); daylight image FLIR _05011 (**b**); nightlight image FLIR _05697 (**c**); and nightlight image FLIR_08926 (**d**).

An additional set of 42 new, diverse images was defined as the Test DS, containing 23 images of nightlight scenes and 19 images captured in daylight. A list of the total datasets used for training the networks is detailed in Table 1. Table 2 lists all Test DS variations that were prepared. Ground truth properties of Test DS are detailed in Appendix A.

The datasets used for network training are shown below.

**Table 1.** Mixed set of images (a total of 2445 daylight and nightlight images) was created with each of the listed processes (IR, pixel-level fusion, RXD anomaly-based fusion, and feature fusion). Datasets containing only day images were separately prepared for RGB, IR, and pixel-level images. DSs containing only nightlight images were prepared for IR images and pixel-level fusion images.

| Dataset | No. of Images | RGB | IR Images | Pixel Level Fusion | RXD Anomaly Fusion | Feature Fusion |
|---|---|---|---|---|---|---|
| Total (Mixed DS) | 2445 | | ✓ | ✓ | ✓ | ✓ |
| Daylight Images DS | 1275 | ✓ | ✓ | ✓ | | |
| Nightlight Images DS | 1170 | | ✓ | ✓ | | |

A list of test datasets is shown below.

**Table 2.** Mixed set of images containing a total of 42 images (19 daylight images and 23 nightlight images) was created with each of the listed processes as the Test DS. RGB images were tested only on daylight images.

| Dataset | No. of Images | RGB Images | IR Images | Pixel Level Fusion | RXD Anomaly Fusion | Feature Fusion |
|---|---|---|---|---|---|---|
| Total (Mixed) Test DS | 42 | | ✓ | ✓ | ✓ | ✓ |
| Daylight Images | 19 | ✓ | | | | |
| Nightlight Images | 23 | | | | | |

### 3.2.3. Post-Processing

The resulting fused sets of images referenced to the original IR images were validated for the most effective process to yield the best physical value as input. Effectiveness in this manner means a robust pre-processing that will reduce the variation between diverse scene image representations while preserving the synergy advantage of the gathered multi-sensor information.

### CNN Installation and Network Training

A convolutional neural network (CNN) based on the YOLO V5 architecture (initially trained on the cityscape dataset) was trained to detect and classify four classes: cars, pedestrians, dogs, and bicycles (the last two classes were later ignored due to a low number of annotations). The model was deployed using the Roboflow framework, installed on PyTorch environment version 1.5, Python 3.7, and CUDA 10.2., and was executed with Google Colab, which facilitates a 12 GB NVIDIA Tesla K80 GPU. The YOLO V5 structure is presented in Figure 9. The input size was set to 640, and the batch size to 16. The training was set to 750 epochs. Data allocation was set to 70% for training, 20% for validation, and the remaining 10% for tests. The initial learning rate was set to 0.01. YOLO V5 was chosen for training for the advantages mentioned in Section 1, as well as for it being smaller and generally easier to use in production, and the model's eligible image input types, together with the connectivity offered by the model between its platform and a free storage framework. Another advantage is the free access the model offers for training multiple networks.

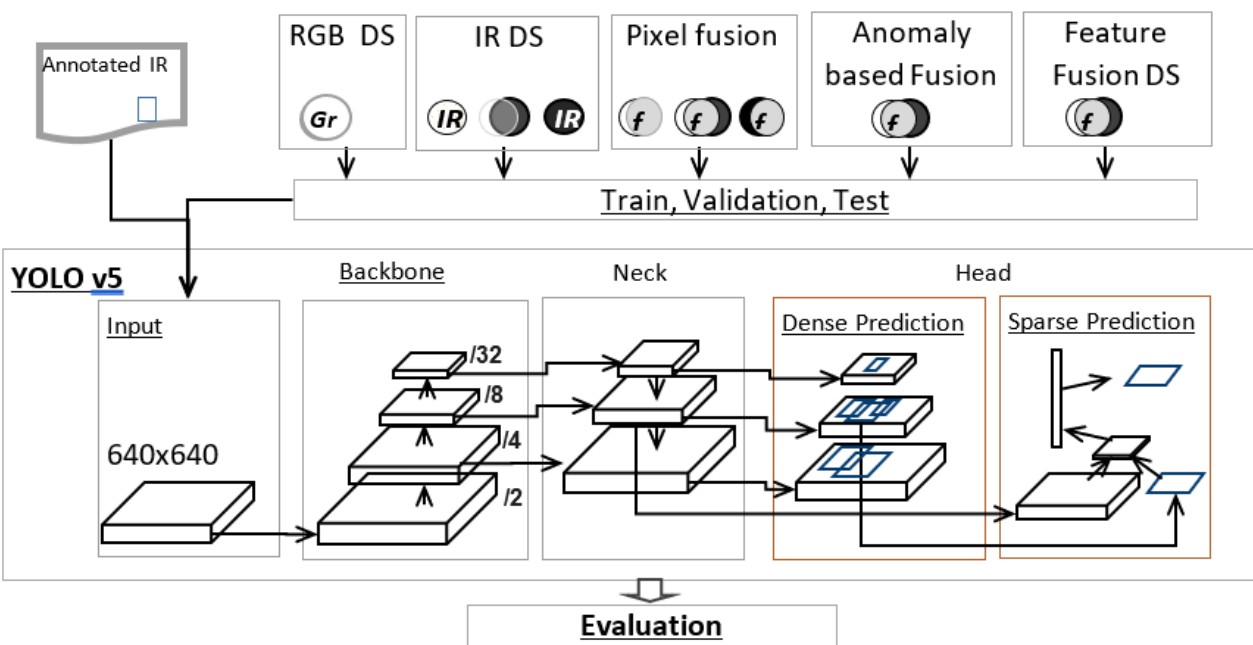

**Figure 9.** Post-process phase where each prepared training DS is being fed as a YOLO V5 model input, with the IR annotated images as GT to train the network.

At first, the network was separately trained with each of the daylight/nightlight sets: RGB images (daylight dataset only) and IR images, and low-level processed images that were fused at the pixel level for comparison.

Next, the network was trained with the mixed datasets (daylight and nightlight images) of IR images, RXD Anomaly Fusion, feature fusion, and pixel-level fusion. Weights of each trained network were used to detect and classify the suitable Test DS.

Detected objects were marked with bounding boxes (BB) and labeled with suitable label class names. Results were evaluated using a confusion matrix (pixel level) according to the following indicators (Equations (5)–(7)):

$$Precision = T_d/GT \tag{5}$$

where $T_d$ is the number of truly detected pixels divided by *GT*, which stands for the total number of ground truth pixels (overlap between YOLO and ground truth), and

$$Recall = OA_d/GT, \tag{6}$$

in which $OA_d$ is the number of all detected pixels by YOLO divided by *GT*.

$$IOU = T_d/(GT + OA_d - T_d) \tag{7}$$

The IoU (Intersect over Union) indicator is calculated by dividing $T_d$ (that is, the number of true detected pixels by YOLO) with the sum of *GT* and $OA_d$ minus $T_d$, reflecting the number of pixels in all marked areas (both in GT and in YOLO, overlapping pixels counts once).

An image is considered successfully classified when gained IoU > 0.5.

The networks' prediction performances were compared using the F1 score (Equation (8)):

$$F1score = 2(Precision \times Recall)/(Precision + Recall) \tag{8}$$

## 4. Results

The training results with different datasets were examined for classes (cars and pedestrians) in daylight and nightlight images according to the classification scores (average IoU).

Given the above-mentioned challenges of image complexity, the proposed pre-processing was applied before feature fusion in an attempt to create a feature-based dataset using different threshold ranges for each of the source images. An additional effect that seems to decrease the efficiency of feature fusion according to a level of intensity threshold is the patchiness of the gathered artificial input caused by integrating layers of data, which create artificial edges that might overcome the object's edges. These limitations ended up in partially segmented, noisy, fused images. Therefore, intensity-based feature fusion was not tested further in this framework.

Tables 3 and 4 detail the networks' scores in classifying the unseen test dataset images, which were processed with the same method each network was trained.

Training the network to detect cars with only daylight images (Table 3) yields the best result in the pixel-level fusion method (83%), slightly better than the IR dataset (81%) and much better than classifying cars in daylight scenes using RGB unprocessed dataset (68%). Networks trained with only nightlight images (Table 4) result in a correct classification of 82% using the IR dataset and only 74% when trained on the pixel-level fusion dataset.

**Table 3.** IoU classification scores of networks trained with only daylight images. Different types of images (RGB, IR, and fused) were used in each training process. Results specify the average IoU scores in detecting and correctly classifying cars and pedestrians by the trained network compared to GT annotations.

| Daylight Dataset: Average IoU | | | |
|---|---|---|---|
| Class | RGB | IR | Pixel Fusion |
| Cars | 68% | 81% | 83% |
| Pedestrians | 35% | 53% | 47% |

**Table 4.** IoU classification scores of networks trained with only nightlight images. Different types of images (IR and fused) was used for each training process. Results specify the average IoU scores in detecting and correctly classifying cars and pedestrians by the trained network compared to GT annotations.

| Nightlight Dataset: Average IoU | | | |
|---|---|---|---|
| Class | RGB | IR | Pixel Fusion |
| Cars | - | 82% | 74% |
| Pedestrians | - | 71% | 54% |

Training networks with a mixed dataset (Table 5) show an improvement in car detection (84% success) compared with training networks using IR daylight images separately. Training with a mixed dataset contributes to the same scores (84%) in all fusion methods that were tested for classifying cars in daylight images. All networks trained with fused mixed datasets show good performance in classifying cars (above 80%) in nightlight images. Pixel-level fusion reached the best score of 84% for correct classification.

None of the methods overperformed the 84%, hinting at the IoU benchmark drawbacks to be discussed further.

**Table 5.** IoU scores for cars' classification of networks trained with Mixed DS of the different processes. Average IoU score is for all images in the dataset; average day and average night separately sum the relative score for daylight images and for nightlight images out of the mixed DS, respectively. Scores are calculated in relation to GT annotations.

| | Mixed Dataset: Cars | | | |
|---|---|---|---|---|
| Class: Cars | RXD An. Fusion | Feature Fusion | Pixel Fusion | IR |
| Average IoU | 82% | 82% | 84% | 84% |
| Average Day | 84% | 84% | 84% | 84% |
| Average Night | 81% | 80% | 84% | 84% |

Detection and classification of pedestrians, however, is distributed in a wider range of scores and can imply the challenges of this task. Early training attempts using separated datasets for daylight and nightlight images led to low performance in all datasets: the unprocessed IR dataset reached 53%, followed by 47%, and 35% with the pixel-level fusion dataset and unprocessed RGB image, respectively (Table 3). Training networks with mixed datasets (Table 6) showed improved results in classifying pedestrians in daylight images. The best scores were reached with the IR dataset (68%), followed by feature fusion-level and anomaly-level fusion methods that also reached relatively high scores (64% and 62%, respectively). The lowest score for classifying pedestrians in daylight images is shown when the network was trained with a mixed dataset at pixel-level fusion (55%).

Training the networks with a mixed DS: pedestrians

**Table 6.** IoU scores for pedestrians' classification by networks trained with Mixed DS of the different processes compared to IR scores. Average IoU score is of all images in the datasets; average day and average night separately sum the relative score for daylight images and for nightlight images out of the mixed DS, respectively. IoU scores are calculated in relation to GT annotations.

| | Mixed Dataset: Pedestrians | | | |
|---|---|---|---|---|
| Class: Pedestrians | RXD An. Fusion | Feature Fusion | Pixel Fusion | IR |
| **Average IoU** | 75% | 72% | 65% | 73% |
| **Average Day** | 62% | 64% | 55% | 68% |
| **Average Night** | 81% | 79% | 75% | 78% |

Networks trained with mixed DS show the advantage of the proposed anomaly-level fusion method in classifying pedestrians in nightlight scenes (Table 6). Using the proposed method achieves 81% correct pedestrian classification in nightlight, an improvement compared with the other tested datasets (79%, 78%, and 75% with feature-level fusion, IR, and pixel-level fusion, respectively) and a significant improvement over training with a dataset of nightlight images only, which yielded 71% on IR dataset and 54% using the pixel-level fusion dataset (Table 4).

An IoU score higher than 0.5 was set as an indication of the network's success in classifying an image. The number of images each network failed to classify was counted. Results of networks' failure in classifying objects to the selected categories are summarized in Tables 7 and 8 (detailed tables appear in Appendix B) according to the dataset type and classification category.

**Table 7.** Number of images out of Test DS images in which the trained networks failed to classify car objects, with IoU detection rate lower than 0.5. Columns: DSs used to train the network (i.e., RGB images (daylight dataset only), IR images, and pixel-level fused images). Rows: the actual part of the dataset the network was trained with (i.e., Daylight dataset, Nightlight dataset and Mixed dataset).

| *Cars* | | RGB | IR | Pixel Fusion |
|---|---|---|---|---|
| Daylight Dataset | | 1 | 1 | 3 |
| Nightlight Dataset | | | 2 | 3 |
| | **Total IoU < 0.5** | | 3 | 6 |
| 38 | FN rate | | 8% | 16% |
| Mixed Dataset | **Total IoU < 0.5** | | 1 | 0 |
| 38 | FN rate | | 3% | 0% |

**Table 8.** Number of images out of Test DS images in which the trained networks failed to classify pedestrians with IoU detection rate higher than 0.5. Columns: DSs used to train the networks (i.e., RGB images, IR images, and pixel-level fused images). Rows: actual part of the dataset the network was trained with (i.e., Daylight dataset, Nightlight dataset and Mixed dataset).

| *Pedestrians* | | RGB | IR | Pixel Fusion |
|---|---|---|---|---|
| Daylight Dataset | | 13 | 7 | 8 |
| Nightlight Dataset | | | 4 | 11 |
| | **Total IoU < 0.5** | | 11 | 19 |
| 34 | FN rate | | 32% | 56% |
| Mixed Dataset | **Total IoU < 0.5** | | 4 | 8 |
| 34 | FN rate | | 12% | 24% |

A network trained with a mixed dataset of IR images failed to classify cars in one image. A network trained with pixel-level fusion succeeded in classifying all images as car objects. The feature-level fusion and the anomaly-level fusion images did not use daylight and nightlight images as separate datasets for training; therefore, it is not included in this comparison.

## 5. Discussion

The results show an advantage for training mixed datasets over separated datasets for daylight and nightlight images. A key factor for this analysis is that none of the methods overperformed (84% IoU) for pedestrian classification. This limit might be caused by the disadvantage of the evaluation method.

The fusion of multiple sensors can yield synergy in fused data by preserving valuable information from each data source. The distinct advantage of the IR sensor is the ability to expose warmer objects (e.g., pedestrians) in a dark scene and its insensitivity to reflected dazzling lights. The IR image is usually flattened relative to RGB images and lacks object depth information.

IoU is a known and acceptable benchmark for measuring neural network performance in segmentation and classification tasks. However, this benchmark has several limitations in performance evaluation. First, IoU is calculated based on the ground truth, and the benchmark score is influenced by the annotation quality. An unmarked object (by an annotator) that is classified by the network is considered an error, while often, these detected objects are correct, so the network classifies better than the annotator. In addition, the proximity of dynamic objects to the scanning sensors and their position (object's size as a proportion of the image size) may affect the detection scores differently. Naturally, closer objects are marked with a large area (number of pixels for bounding box), and far objects are marked with a smaller bounding box size. A minor shift in a bounding box created by the network, relative to the marked ground truth, may differently affect the

measured error ratio on bounding boxes with a small number of pixels (the number of counted pixels in a shifted bounding box area to bounding box size is comparatively high), for which any shift in the bounding box location might slightly reduce the score. Hence, the IoU benchmark was used to evaluate the network's success in segmentation and classification tasks.

Examination of the contribution of a mixed dataset to the success of the network in classifying cars from daylight images revealed that only three images were detected with IoU scores lower than 50%. The total average for detecting cars in daylight and nightlight is 82.7%, and the distributed averages are 83.7% for daylight and 81.8% for nightlight images. Some of the lowest scores were caused by a greater accuracy of the network in object detection relative to the annotated ground truth, as will be detailed later.

A network trained on a mixed dataset, processed with anomaly-based fusion, succeeded in the classification of all daylight images. The overlap rate between network detection to the annotated ground truth reached 89%, and the area marked by the network to the annotated ground truth area was 96.5%, meaning a high accuracy level for classifying cars in daylight images. The image in Figure 10 shows the effect caused by different resolutions in the dual image: part of the ground truth annotation marked on the IR image is out of the frame captured by the RGB image. Fusion enables us to overcome these challenges. Furthermore, unmarked cars in the manual ground truth annotation were detected and correctly classified by the network, leading to an IoU score of 73% for this image, which might be considered a false negative from the IoU score.

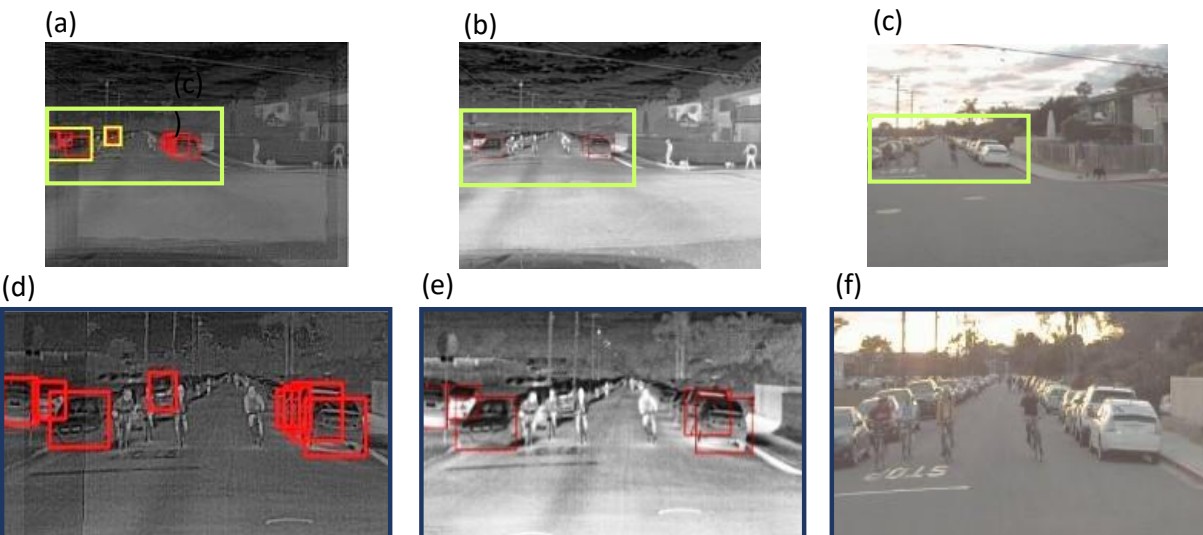

**Figure 10.** Image FLIR_06423: example of dual images multi-resolution effects on classifying cars in day images. A subsection marked in network classification of the RXD Anomaly Fusion image (**a**), GT image (**b**), and original RGB image (**c**), with light green boxes that are zoomed in to the region of interest, shown in images (**d**,**e**,**f**), respectively. The three marked BBs on the left of the GT (b) are out of the frame of the RGB image (**c**). The proposed pre-processing enables the network to detect these objects correctly.

A network trained on a mixed dataset of anomaly-level fused images classified cars in all nightlight images of the test dataset. The overlap ratio between the marked area by the network to the annotated ground truth was 91.8%. The networks' classified area to the annotated ground truth area ratio was 110%. These scores show the network's high accuracy level, though the error rate is slightly increased. Some of this erroneous rate is due to correct networks' over-classification of unmarked objects in the annotated ground truth (Figure 11).

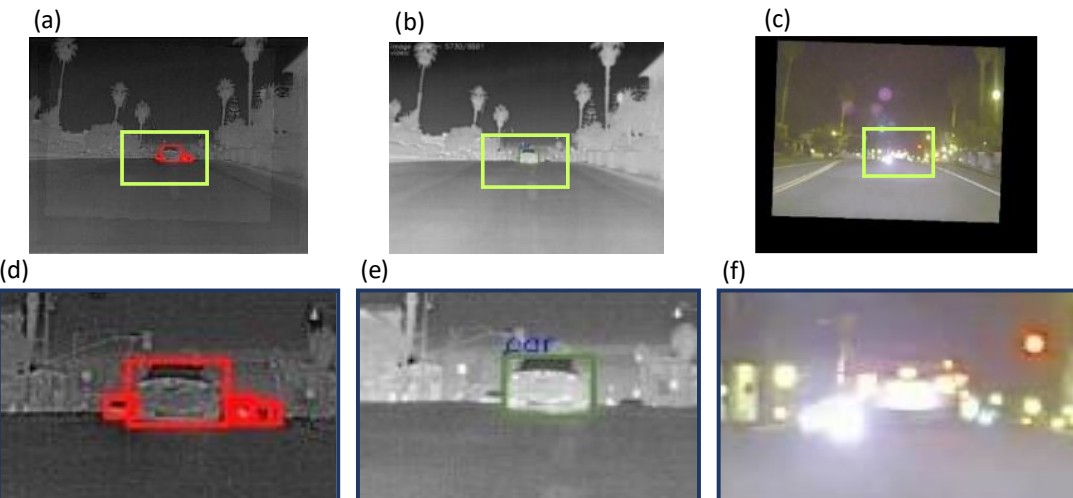

**Figure 11.** Image FLIR_05731: low score example of correct car classification in a night image. A subsection marked in: network classification of the RXD Anomaly Fusion image (**a**), GT image (**b**), and original RGB image (**c**), with light green boxes that are zoomed in to the region of interest, shown in images (**d**,**e**,**f**), respectively. Car objects that were detected by the network (**a**,**d**) are unmarked objects in GT (**b**,**e**).

A network trained on a mixed dataset of anomaly-level fused images reached an average IoU of 62.6% in classifying pedestrians in daylight images. The average overlap ratio between the network's classification area to the annotated ground truth area was 70.1%. The average network's classified area to the annotated ground truth area was 88.3%.

In some images, few objects were misclassified by the network, yet several images gained low scores due to pedestrian classification based on the correct detection of unmarked ground truth annotation (Figure 12).

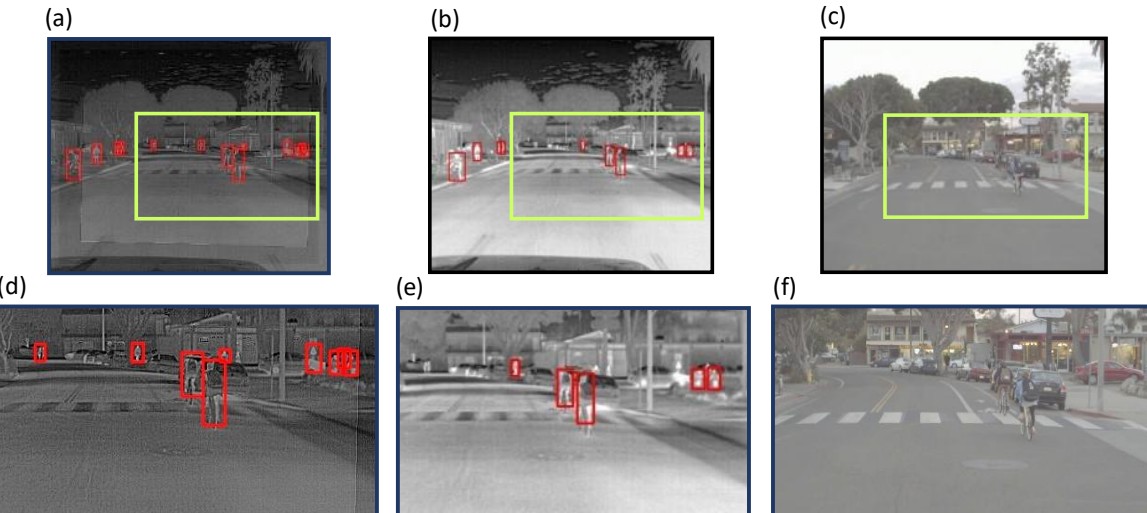

**Figure 12.** Image FLIR_08807: an example of pedestrian classification based on the correct detection of unmarked ground truth annotation in a day image by the network (**a**,**d**) vs. unmarked pedestrians in GT annotation (**b**,**e**). A subsection marked in network classification of the RXD Anomaly Fusion image (**a**), GT image (**b**), and original RGB image (**c**), with light green boxes that are zoomed in to the region of interest, shown in images (**d**,**e**,**f**), respectively.

The annotated ground truth area ratio was 111%. Out of two images that scored less than 50% IoU, the first was partially classified by the network, while the second was correctly classified by the network as a missing annotation on ground truth.

An additional drawback of the IoU marker (Figure 13) arises when classifying small (usually far) objects. A minor shift in the detected bounding box from the annotated ground truth bounding box leads to a substantial reduction in the IoU score, despite correct classification by the network. Image FLIR_07989, for example, was successfully classified by the network but scored only 71%.

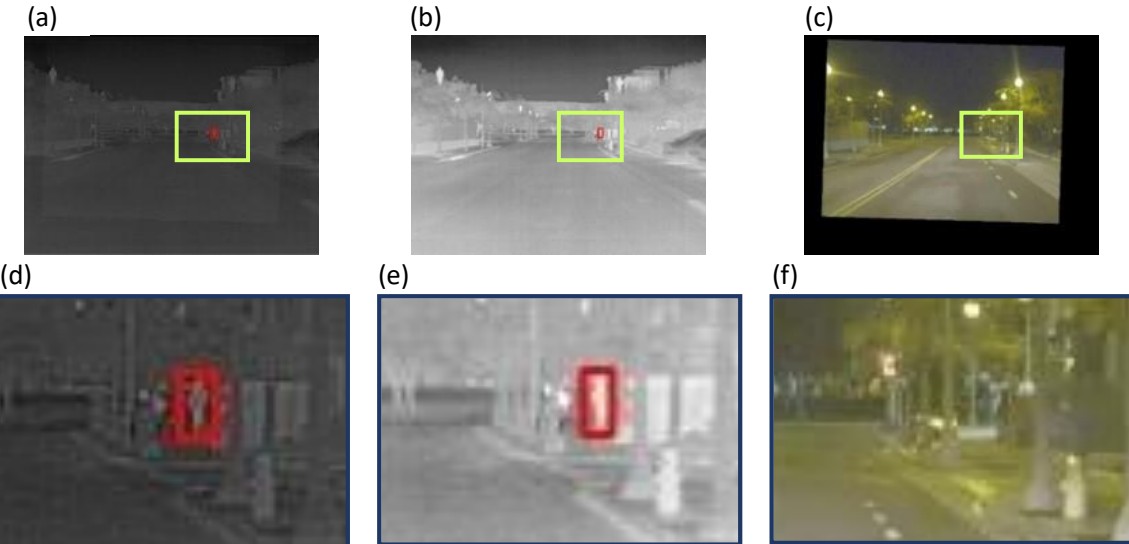

**Figure 13.** Image FLIR_07989: example of the bounding box size effect on IoU score. Despite the correct detection and classification, this image's IoU score is 71%. A subsection marked in network classification of the RXD Anomaly Fusion image (**a**), GT image (**b**), and original RGB image (**c**), with light green boxes that are zoomed in to the region of interest, shown in images (**d**), (**e**), and (**f**), respectively.

Each trained network's result was separately summed up for daylight and nightlight datasets classification in comparison to the mixed dataset classification to evaluate the contribution of mixed datasets in lowering the network classifications' failure. This evaluation was conducted based on a trained network with IR images comparing pixel-level fusion images only.

Images containing zero objects from one of the classes (overall GT area = 0) were eliminated due to computation limitations (divided by zero). In the car category, out of 42 images in the test set, four images with no cars were eliminated. The network performance was tested on 38 mixed images, out of which 17 were in daylight, with the remaining 21 in nightlight. In the pedestrian category, eight out of 42 images were discarded (images with zero objects in the pedestrians' category). In practice, the network's performances in classifying pedestrians were tested on 34 images, out of which 16 images were in daylight and 18 images were in nightlight.

The network trained with IR images on separate datasets for daylight and nightlight (Table 5) failed to classify cars in three images in total (8%). A network trained on a mixed dataset decreased the rate of failure in car classification to one image only (3%), an improvement of 5% in the network's success.

Training the network with separate datasets of IR images for daylight and nightlight (Table 6) resulted in 32% in pedestrian classification (a total of 11 misclassified images). Training on a mixed dataset decreased the rate of failure of the network to 12%, a total of four out of 34 images.

The network trained with pixel-level fused images on separate datasets for daylight and nightlight failed to classify six images with car objects (16% out of 34 images with car annotations). A network trained on a mixed dataset successfully classified cars in all images with an IoU score higher than 50%, i.e., 0% network failure (Table 5).

Training the network with separate datasets of pixel-level fused images failed in 56% of pedestrian classifications (a total of 19 misclassified images). Training on a mixed dataset decreased the rate to 24%, a total of eight images out of 34 daylight and nightlight mixed scenes and improved the network's success by 32% in classifying objects as pedestrians (Table 6).

## 6. Conclusions

The pre-processing method intends to handle issues emerging from combining input from multiple types of sensors, such as data registration, values unification, and statistical-based anomaly detection for foreground refinement. By doing so, a reduction in the amount of the gathered data and its variation level caused by differences in sensor types and properties, different lighting and environmental conditions, complex scenes, dynamic objects, etc., was achieved. The unified physical value contributed to the robustness of input data extraction, hence obtaining a better perception of the surroundings under varied environmental states.

We applied the anomaly-level fusion method to suppress the effects of complex dynamic background captured by moving cameras, enabling the model to concentrate on the spatial variation of the moving foreground. With the differences in fusion processes and their contribution to car and pedestrian classification, it can be seen that, due to cars' flat and smooth textures, all proposed fusion processes yielded a high detection score (80-84%) in classifying a mixed dataset (of daylight and nightlight images) after eliminating the masked effects. Thus, the fusion process, in the context of car detection, makes a major contribution to suppressing masking and background effects.

As for pedestrian detection, a network trained with a mixed dataset of RXD anomaly-level fused images gains the highest average IoU score (75%). The bright appearance of the objects in the IR images contributes to the object's flattening and, as a result, separately accentuates it from its background. A CNN network trained with RXD anomaly-level fused images classifies pedestrians in nightlight images best (81%) by suppressing the background and masking effects, thus lowering the dependency on accurate registration while maintaining the brightness of pedestrians' appearances.

Classifying pedestrians from a complex background in daylight images, however, appears to be the most challenging task for the networks, a category that results in the lowest scores in all proposed processes. Enriching the object with texture and depth limits the network's ability to classify pedestrians as a united, single object; hence, the thermal images achieve the best scores (68%) in pedestrian classification in daylight images. RF feature-level fusion slightly increases the pedestrian's gradient and adds no texture, while fusing with the IR image yields 64%, slightly lower than IR images and the best out of the examined fusion processes. The authors concluded that as the fusion process expresses a greater range of details from the visual image, the network's IoU scores in classifying pedestrians in daylight images decrease.

This understanding reinforces the benefits of expressing the color image as a physical value that can contribute to the robustness of the network in training mixed datasets (daylight and nightlight images) by moderating the range of detail enrichment and preserving and neutralizing the masking and background effects in nightlight images.

**Author Contributions:** Conceptualization, A.B.; Methodology, R.B.-S. and A.B.; Software, R.B.-S.; Validation, R.B.-S.; Formal analysis, R.B.-S.; Investigation, R.B.-S. and A.B.; Data curation, R.B.-S.; Writing – original draft, R.B.-S.; Writing – review & editing, A.B.; Supervision, A.B.; Project administration, A.B.; Funding acquisition, A.B.

**Funding:** This research was supported by grants (no. 67437 and 74538) from the Israel Innovation Authority's AVATAR consortium (Autonomous Vehicle Advanced Technologies for situational AwaReness).

**Data Availability Statement:** Not applicable.

**Acknowledgments:** The authors of this paper gratefully acknowledge Third-Eye Systems LTD. For their fruitful collaboration. A sincere appreciation is expressed to Erez Nur, the technical manager of the consortium. Special thanks to Shay Silberklang and Alexander Logovinsky for their assistance in code writing and designing, technical consultancy, and their friendly attitude. Special thanks to Gabriel Cotlier for his kind advice in English editing.

**Conflicts of Interest:** The authors declare no conflicts of interest. The funders had no role in the design of the study; in the collection, analyses, or interpretation of data; in the writing of the manuscript; or in the decision to publish the results.

## Appendix A. List of Test Dataset Images

List of the test dataset images split into daylight images (Table A1) and nightlight images (Table A2). The number of annotated cars and pedestrians per image is specified, in addition to the total BB area per class, in pixels (overlapping BB pixels were counted once).

**Table A1.** Day images, Test DS.

| Image Name | Number of Cars BB (Cars GT) | Overall GT BB Area (in Pixels) | Number of People BB (People GT) | Overall GT BB Area (in Pixels) |
|---|---|---|---|---|
| FLIR_06345 | 9 | 26,350 | 9 | 6388 |
| FLIR_06423 | 5 | 8461 | 6 | 8833 |
| FLIR_08653 | 8 | 11,454 | 0 | 0 |
| FLIR_08692 | 2 | 1266 | 1 | 1085 |
| FLIR_08727 | 6 | 5607 | 4 | 3598 |
| FLIR_08807 | 7 | 7351 | 9 | 7810 |
| FLIR_08821 | 6 | 45,497 | 5 | 2938 |
| FLIR_08834 | 3 | 6666 | 6 | 7151 |
| FLIR_09086 | 3 | 14,492 | 9 | 11,526 |
| FLIR_09118 | 2 | 23,521 | 7 | 16,468 |
| FLIR_09231 | 4 | 16,341 | 0 | 0 |
| FLIR_09276 | 4 | 11,037 | 4 | 1261 |
| FLIR_09282 | 4 | 58,705 | 2 | 8226 |
| FLIR_09296 | 0 | 0 | 6 | 51,952 |
| FLIR_09929 | 4 | 5106 | 3 | 4930 |
| FLIR_10013 | 0 | 0 | 0 | 0 |
| FLIR_10094 | 8 | 12,538 | 4 | 33,398 |
| FLIR_10119 | 4 | 21,615 | 4 | 3662 |
| FLIR_10121 | 3 | 9912 | 4 | 5224 |

**Table A2.** Nightlight images, Test DS.

| Image Name | Number of Cars BB (Cars GT) | Overall GT BB Area (in Pixels) | Number of People BB (People GT) | Overall GT BB Area (in Pixels) |
|---|---|---|---|---|
| FLIR_05731 | 1 | 1945 | 0 | 0 |
| FLIR_05737 | 1 | 2347 | 6 | 1920 |
| FLIR_05849 | 4 | 11,032 | 2 | 779 |
| FLIR_05938 | 3 | 1735 | 2 | 3046 |
| FLIR_06001 | 0 | 0 | 5 | 1685 |
| FLIR_06040 | 1 | 236 | 2 | 598 |
| FLIR_07495 | 7 | 11,690 | 2 | 5986 |
| FLIR_07498 | 6 | 4785 | 2 | 8299 |
| FLIR_07500 | 5 | 4029 | 2 | 772 |
| FLIR_07986 | 17 | 38,681 | 0 | 0 |
| FLIR_07989 | 2 | 8583 | 1 | 296 |
| FLIR_08079 | 5 | 4824 | 3 | 3293 |
| FLIR_08086 | 3 | 3518 | 4 | 2208 |
| FLIR_08087 | 4 | 20,213 | 5 | 1197 |
| FLIR_08237 | 4 | 10,719 | 10 | 5604 |
| FLIR_08318 | 5 | 13,006 | 12 | 13,278 |
| FLIR_08522 | 6 | 14,717 | 0 | 0 |
| FLIR_08523 | 4 | 5357 | 1 | 476 |
| FLIR_08527 | 0 | 0 | 0 | 0 |
| FLIR_08926 | 5 | 12,754 | 4 | 12,704 |
| FLIR_08932 | 5 | 11,306 | 8 | 37,928 |
| FLIR_08954 | 5 | 30,679 | 1 | 6294 |
| FLIR_09669 | 8 | 26,896 | 0 | 0 |

## Appendix B

### Training networks with separated DS vs. mixed DS: list of images with low IoU classification scores (<0.5)

<u>Training networks with separated datasets for day and night: images with IoU < 0.5 in the cars' classification</u>

**Table A3:** Networks trained with daylight image datasets failed to classify 1-3 images for car objects, out of 17 day images

| Daylight Dataset: Cars | | | | | | |
|---|---|---|---|---|---|---|
| Time | Image Number | Image Name | BB Count in GT | RGB Day | IR Day | Pixel Fusion Day | Fails Per Image |
| **Day** | 22 | FLIR_08653 | 8 | - | | | 1 |
| **Day** | 40 | FLIR_10094 | 8 | | - | - | 2 |
| **Day** | 41 | FLIR_10119 | 4 | | | - | 1 |
| **Day** | 42 | FLIR_10121 | 3 | | | - | 1 |
| **17** | | | | 1 | 1 | 3 | |

**Table A4.** Networks trained with nightlight image datasets failed to classify 2–3 images for car objects out of 21 night images.

| | | Nightlight Dataset: Cars | | | | |
|---|---|---|---|---|---|---|
| Time | Image Number | Image Name | BB in GT | IR Night | Pixel Fusion Night | Fails Per Image |
| **Night** | 3 | FLIR_05849 | 4 | | - | 1 |
| **Night** | 4 | FLIR_05938 | 3 | - | - | 2 |
| **Night** | 20 | FLIR_08523 | 4 | - | - | 2 |
| **21** | | | | 2 | 3 | |

Training networks with separated datasets for day and night: images with IoU < 0.5 in the pedestrians' classification

**Table A5.** Networks trained with day image datasets failed to classify 7–13 images of pedestrian objects out of 16 day images.

| | | Daylight Dataset: Pedestrians | | | | | |
|---|---|---|---|---|---|---|---|
| Time | Image Number | Image Name | BB Count in GT | RGB Day | IR Day | Pixel Fusion Day | Fails Per Image |
| **Day** | 7 | FLIR_06345 | 9 | - | | | 1 |
| **Day** | 25 | FLIR_08807 | 9 | - | | | 1 |
| **Day** | 26 | FLIR_08821 | 5 | - | | | 1 |
| **Day** | 27 | FLIR_08834 | 6 | - | | | 1 |
| **Day** | 31 | FLIR_09086 | 9 | - | - | - | 3 |
| **Day** | 32 | FLIR_09118 | 7 | - | - | - | 3 |
| **Day** | 34 | FLIR_09276 | 4 | - | - | - | 3 |
| **Day** | 35 | FLIR_09282 | 2 | - | - | - | 3 |
| **Day** | 36 | FLIR_09296 | 6 | - | | | 1 |
| **Day** | 38 | FLIR_09929 | 3 | - | | - | 2 |
| **Day** | 40 | FLIR_10094 | 4 | - | - | - | 3 |
| **Day** | 41 | FLIR_10119 | 4 | - | - | - | 3 |
| **Day** | 42 | FLIR_10121 | 4 | - | - | - | 3 |
| **16** | | | | 13 | 7 | 8 | |

**Table A6.** Networks trained with night image datasets failed to classify 4–11 images for pedestrian objects out of 18 night images.

| | | Nightlight Dataset-Pedestrians | | | | |
|---|---|---|---|---|---|---|
| Time | Image Number | Image Name | BB in Gt | IR Night | Pixel Fusion Night | Fails Per Image |
| **Night** | 3 | FLIR_05849 | 2 | - | - | 2 |
| **Night** | 6 | FLIR_06040 | 2 | | - | 1 |
| **Night** | 11 | FLIR_07500 | 2 | - | - | 2 |
| **Night** | 13 | FLIR_07989 | 1 | | - | 1 |
| **Night** | 15 | FLIR_08086 | 4 | | - | 1 |
| **Night** | 16 | FLIR_08087 | 5 | | - | 1 |
| **Night** | 17 | FLIR_08237 | 10 | | - | 1 |
| **Night** | 20 | FLIR_08523 | 1 | - | - | 2 |
| **Night** | 28 | FLIR_08926 | 4 | - | - | 2 |
| **Night** | 29 | FLIR_08932 | 8 | | - | 1 |
| **18** | | | | 4 | 11 | |

Training networks with Mixed datasets: images with IoU < 0.5 in cars' classification and pedestrian's classification

**Table A7.** Networks trained with mixed datasets failed to classify a single image for car objects, out of 38 day and night images.

| Mixed Dataset: Cars | | | | | | |
|---|---|---|---|---|---|---|
| Time | Number | Image Name | BB in GT | Tir Mix | Pixel Fusion Mix | Fails Per Image |
| **Night** | 4 | FLIR_05938 | 3 | - | | 1 |
| **38** | | | | 1 | 0 | |

**Table A8.** Networks trained with mixed datasets failed to classify 4–8 images for pedestrian objects out of 34 day and night test images.

| Mixed Dataset-Pedestrians | | | | | | |
|---|---|---|---|---|---|---|
| Time | Number | Image Name | BB in GT | Tir Mix | Pixel Fusion Mix | Fails Per Image |
| **Night** | 3 | FLIR_05849 | 2 | | - | 1 |
| **Night** | 11 | FLIR_07500 | 2 | - | - | 2 |
| **Night** | 28 | FLIR_08926 | 4 | - | - | 2 |
| **Day** | 31 | FLIR_09086 | 9 | | - | 1 |
| **Day** | 34 | FLIR_09276 | 4 | - | - | 2 |
| **Day** | 35 | FLIR_09282 | 2 | | - | 1 |
| **Day** | 40 | FLIR_10094 | 4 | | - | 1 |
| **Day** | 41 | FLIR_10119 | 4 | - | - | 2 |
| **34** | | | | 4 | 8 | |

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
