# Peer review of "Fused Thermal and RGB Imagery for Robust Detection and Classification of Dynamic Objects in Mixed Datasets via Pre-Trained High-Level CNN"

_remotesensing, doi:10.3390/rs15030723_

Round 1

Reviewer 1 Report

The authors propose a multi-sensorial data pre-processing method for handling issues emerging from combining input from multiple types of sensors, such as data registration and values unification.

Unfortunately, the paper contains the following problems.

    • The paper language needs polishing since it contains syntax errors.

    • The paper contains a number of citations that need correction (e.g. in the first paragraph on page 3: [20] proposed to fuse… should change to Li and Wu [20] proposed to fuse …). 

    • Subsection 1.1 must become a separate main section.

    • At the end of the “Introduction” section, a small paragraph explaining the paper’s structure must be added.

    • In subsection 1.1, a), b) and c) must be in the same paragraph.

    • All figures inside the text must contain a small paragraph explaining them. This paragraph must be placed next to each figure’s caption.

    • Subsection 2.2.2 must move to the next page.

    • Figure 4 must move to page 9. 

    • Each figure must be referenced as “Fig. x” and not as “Image x”, where x is the figure’s number.

    • The caption in figure 5 must be placed under the figure and not on a new page.

    • Subsection 2.3.2 must move to the next page.

    • The authors must include a separate section explaining the Yolo V5 convolutional neural network (CNN) architecture they have used. A diagram of its architecture must accompany the explanation of this CNN.

    • In the experimental part, the authors must include a table summarizing the parameters used for the execution of the experiments.

    • Page 26 is blank and must be removed.

Author Response

The authors greatly appreciated your constructive and helpful comments/suggestions. We have included below a point-by-point response to the raised concerns. We believe that the manuscript is significantly improved after this revision. Additionally, we have tracked the changes in the revised manuscript.

Enclosed please find a copy of our notes. 

Reviewer 2 Report

Dear Editor and Dear Authors,

The topic of this paper is very interesting and thank you for your trust to send me the manuscript to review it.

The manuscript proposes a pre-processing method of multi-sensorial data from RGB and thermal cameras to handle issues emerging from combining input from multiple types of sensors such as data registration and values unification.

The comments and suggestions to authors can be found below:

1.      The paper contains Personal Pronouns (we, our,...) that are unusual and not acceptable for the scientific English. The authors should use Passive Voice instead, since that is the standard for scientific English in high quality journals.

2.      The Abstract is not bad, however it is unusual to use abbreviations and text in brackets in it. It is very repulsive and hard to read. It should be corrected. The abbreviations should be explained later in text. Also there are typing mistakes in Abstract.

3.      The Introduction section is good, however a brief summary of the paper should be given in the end.

4.      The Related Work is satisfactory, but it should be placed in a separate section.

5.      I think the Dataset sub-section should be placed in Result section. Also, some parts of the text are not edited well. Authors should check the journal’s template.

6.      The images in figures should be enlarged maximally to increase their visibility, since the paper relies on them in analysis and method explanation. Further, the captions of the figures are not appropriate everywhere, and somewhere the images are damaged (looks like they were copied from other work). Also, the fonts in the text are different in some part!!!

7.      In Fig. 3, the title of the block is missing in block diagram. This is a huge omission, since the block diagram should reflect effectively the research itself.

8.      In page 7, after the expression (1) the text is not clear! It should be the steps of an algorithm, or some sub-algorithm? It is not clarified in the text, thus it is hard to understand what is the aim of it.

9.      The registration sub-section is not clear too. The R, G, B three channels were registered, or, these channels were registered with IR image? Please, rewrite this sub-section with clarified and detail explanation. What registration method was used?

10.  The image fusions part is very hazy and it is very hard to understand what would like to introduce the Authors. It should be reorganized and clarified what method was used.

11.  In 2.3.1. sub-section was first mentioned the anomaly detection, later it is explained in other 2.3.2. sub-section? It is hard to follow. Also, in page 12 at the beginning there is again a explanation of some operation steps? These steps refers to some sub-algorithm? It should be clarified. The parameters in equation (5) are not explained well.

12.  Overall, the proposed method is written very hazy, it is hard to follow and it should be reorganized and rewritten to meet the appropriate quality of the presentation.

13.  In Results section the first sub-section is not numbered (I think it is a sub-section title, please check it). The result and the tables should be explained with more details.

14.  The images in Discussion should be enlarged. It is difficult to notice the marked objects in the image.

15.  The Conclusion section should me written with concise wording to reflect the research itself.

The paper is very interesting and useful from the practical point of view, however it has drawbacks that should be revised in major.

Recommendation: major revision.

Author Response

(The authors gave the same response as above.)

Reviewer 3 Report

This manuscript presents a pre-processing method of multi-sensorial data (RGB and thermal cameras) in the context of embedded Autonomous Vehicle (AV). The fused image is further fed to a CNN for classification (pedestrian/vehicles). While the context is interesting, the manuscript needs improvement in several sections:

1) Presentation wise I found the manuscript is a bit incomplete, e.g., some Sections are full of irrelevant information, e.g., generally a reader is not interested to read such information in a scientific paper "A YOLOv5 repository was duplicated, and a YAML file was updated to fit the local dataset properties."

This is just an example, the entire manuscript is full of such irrelevant information and is often missing important information.

2) Several important information related to the methodology is missing, e.g., why did you set output to such an irregular size (384x513 resolution). What is the learning rate used in training?

3) I assume coregistering RGB and IR is challenging. It would be interesting to make a complete analysis how misregistration is impacting the output.

4) Why did you chose RXD for anomaly detection and not any other method. More motivation is required.

5) Literature review can be improved, especially regarding RGB/IR fusion. Here are a few suggestions:

i) SiamFT: An RGB-infrared fusion tracking method via fully convolutional siamese networks

ii) Mitigating spatial and spectral differences for change detection using super-resolution and unsupervised learning

iii) Infusion-Net: Inter-and Intra-Weighted Cross-Fusion Network for Multispectral Object Detection

iv) Tensor Regression and Image Fusion-Based Change Detection Using Hyperspectral and Multispectral Images

Author Response

(The authors gave the same response as above.)

Round 2

Reviewer 1 Report

The authors have addressed my comments.

Author Response

Thank you, the paper was revised and corrected

Reviewer 2 Report

Dear Editor and Dear Authors,

The manuscript has been revised and resubmitted. Authors answered all the comments.

The comments and suggestions to authors can be found below:

1.      There are some grammar mistakes, for example in Introduction “static-sate”. Maybe it should be “static-state”?

2.      Thus, the English of the whole paper should be checked again.

The paper is improved and it is ready for publication.

Recommendation: accept.

Author Response

(The authors gave the same response as above.)

Reviewer 3 Report

The authors still need to provide appropriate response to several questions:

1) Not sure why authors think analyzing impact of misregistration is out of scope of this paper.

2) It is still not clear why RXD was chosen for anomaly detection. The answer is vague.

3) Did you process 640*640 image at once or did you split it into patches?

Author Response

Thank you, the paper was revised and corrected

1) Not sure why authors think analyzing impact of misregistration is out of scope of this paper.

Registration process was clarified (page 15 lines 9-17). In addition, Indeed its fascinating topic with great impact on further detection and classification thus we include our deep investigation on it in our next publication which is going to be submitted in the upcoming months

2) It is still not clear why RXD was chosen for anomaly detection. The answer is vague.

We revised other methods as explained in the manuscript and choose to use RXD for its ability to be applied on general image, equally well, on global and local scales. This is particularly important with mixed data (daylight and nightlight) dealing with different illumination conditions such as direct light, local saturations, blurriness, and shades.

3) Did you process 640*640 image at once or did you split it into patches?

The images were used at once.